# The Tournesol dataset:
# Which videos should be more largely recommended?

## Abstract

This paper introduces the Tournesol public dataset, which was collected as part of the online deployed platform `https://tournesol.app`. Our dataset contains a list of 1,116,318 comparative judgments of YouTube videos by 8,804 users of the Tournesol platform. 263,668 of these judgments were about which video should be more largely recommended, while the remaining evaluate secondary criteria like content reliability, topic importance and layman-friendliness. The dataset also exports information about users' pretrust statuses and vouches. It is published at `https://api.tournesol.app/exports/all` under ODC-By license. The data is currently used by Tournesol to make community-driven video content recommendations to over 6,000 users.

## 1 Introduction

Recommendation AIs have become extremely influential. In the last few years, beyond their impacts on mental health [58, 21, 94], because they amplify disinformation, cyberbullying and hate, they have been linked to major geopolitical events, including COVID disinformation [81, 46], the rise of far-right parties [93, 92, 98], and the Rohingya genocides [42, 73]. Crucially, in all these examples, the victims of recommendation AIs are not only their users; hate amplification is threatening entire populations, even when these populations do not use recommendation AIs themselves. This is in sharp contrast with the overwhelming majority of the scientific literature, which assumes that recommendation AIs should be optimized for their users only [1, 71].

As online activities grew, recommendation AIs have *de facto* taken the role that was traditionally played by these intermediate bodies [91, 50]. For instance, by amplifying the cyberbullying of climate scientists, Twitter's AI provoked their exodus from the platform [95], thereby turning climate change into a *mute news*, which is endangering plenty of non-users [3]. The great replacement of the intermediate body by privately owned AIs has been tied to an alarming decline of democratic norms worldwide, as many reports expose a global trend of autocratization [72, 7].

So how do today's large-scale recommendation AIs address the ethical dilemmas that they face billions of times per day, when they are tasked with amplifying some (potentially hateful) content over others (of potential public interest)? Currently, they heavily rely on (highly sophisticated) *machine learning* [26, 65]. In other words, such AIs leverage massive amounts of data to determine which content they will promote at scale. However, as an immediate corollary, such AIs are exposed to *manipulation* by *poisoning* data [89]. In fact, this poisoning has been industrialized, not only by authoritarian states [20, 48], but also by private companies based in the UK [53], Spain [16], Israel [6], France [90] and Switzerland [37]. The magnitude of this industry is well captured by one puzzling statistic: Facebook reportedly removes around *7 billion fake accounts per year* [60].

Submitted to 39th Conference on Neural Information Processing Systems (NeurIPS 2025). Do not distribute.

While a recent line of research has provided numerous poisoning mitigations [15, 34, 35, 30, 83, 76], it is also known that there are fundamental impossibility theorems that prevent accurate learning in highly adversarial, heterogeneous and high-dimensional settings [31, 61, 39, 33]. In particular, *there is no substitute for training datasets of high quality and security*. In particular, to design trustworthy ethical AIs, it is essential to train them on large, secured and trustworthy datasets of human ethical judgments. In this paper, we present the *Tournesol public dataset*, whose goal is to remedy the current state of affairs. More precisely we make the following contributions.

**Contributions.** Our main contribution is to present and share the *Tournesol public dataset*, which can be downloaded directly from `https://api.tournesol.app/exports/all`. The dataset consists of 263,668 pairwise comparisons of the recommendability of 56,796 YouTube video by over 8,804 Tournesol accounts. Additionally, the dataset contains 852,650 pairwise comparisons of the videos' quality on secondary criteria, such as reliability, importance and layman-friendliness. Our dataset, published under ODC-By license, also contains pretrust information about contributors, vouches between contributors, as well as scores computed from the data using SOLIDAGO [14]. Crucially, the dataset was collected in a fully deployed environment with actual stakes, as Tournesol eventually makes recommendations based on the provided data to over 6,000 users.

The paper also presents an analysis of our dataset, with valuable insights for the ethics of content recommendation. One finding is that the topic importance highly matters in Tournesol's contributors' judgments. While caveats apply, this suggests that the attention to "fake news" may be misguided; in fact, the disinformation industry often proceeds *without* producing false information, e.g. by overclaiming positive impacts, shifting blame or bullying critics [77]. Prioritizing greater exposure to *mute news* might be more urgent. Our analysis also highlights the need of psychological-based preference learning models, as we expose biases and variations in contributors' judgments.

Finally, our paper discusses numerous exciting research directions that our public dataset could inspire or facilitate. In particular, we believe that a lot more focus should be given to secure learning under poisoning attacks, but also to *Proof of Personhood*, *expertise validation*, *volition learning*, *active learning* and *resilient collaborative filtering*, among others.

**Literature review.** Tournesol presents a new contribution to the growing field of AI alignment with human values [49, 24, 54, 79], which aims to teach human preferences to AIs, and to design systems that maximize what humans prefer to maximize [84, 56]. Clearly, this requires finding out about humans' judgments on how AIs ought to behave. Unfortunately, so far, to the best of our knowledge, all published content evaluation datasets [52, 12, 78, 100, 101, 97, 23, 8] are consumer-centric, i.e. they report what consumers prefer to consume; not what they regard as recommendable to others.

To collect such data in a realistic setting, Tournesol's dataset draws inspiration from several previous AI ethics solutions, which leveraged *collaborative governance* to address cases of conflictual human judgments. In particular, [64] introduced WeBuildAI, a framework where stakeholders of a food donation system could weigh in on the identity of the recipient of a donation. One challenge is that such decisions must be made every day; but stakeholders are not available every time a decision needs to be made. To account for their preferences, WeBuildAI asks stakeholders to either write down an AI that describes their preferences, or to provide judgments on generated food donation dilemmas. In the latter case, a learning model is then used to infer how the stakeholders would likely assess other dilemmas. In any case, an *algorithmic representative* is thereby constructed for each stakeholder; and the resulting decision will follow from a vote of the algorithmic representatives. Similar approaches were proposed for kidney donation [45] and for the "trolley dilemmas" [43] that autonomous cars could one day face [11, 75].

Perhaps most similar to our approach are Twitter's *Community Notes* [99, 80], whose governance is intended to be fully community-driven. More specifically, the system allows a community of contributors to add a note to misleading tweets, e.g. to correct misinformation or to add context to prevent confusion. The contributors cannot only propose the note; they are also asked to assess other contributors' notes. Notes that are judged helpful by a sufficiently large and diverse set of contributors are then published by the platform. The system is very transparent, and provides a lot of freely accessible data on human judgments[1].

---

[1]The data can be downloaded here: `https://communitynotes.twitter.com/guide/en/under-the-hood/download-data`

**Structure of the paper.** In the sequel, Section 2 will present our public dataset, and the context in which the data was provided. Section 3 presents an analysis of our dataset. Section 4 then provides a list of research challenges that are raised by the dataset. Finally, Section 5 concludes.

## 2 The dataset

In this section, we describe our main contribution, namely the release of a new, scalable, secured and trustworthy database of reliable human judgments.

### 2.1 Raw data

**Pretrust.** To guarantee the security of our data, Tournesol aims to verify that every account is owned and controlled by a human, and that this human only owns and controls this single account on the platform. In other words, Tournesol aims to obtain a *Proof of Personhood* [17] to verify each active Tournesol account, and to thereby prevent *Sybil attacks* [28]. Unfortunately, there is currently no reliable and scalable solution for *Proof of Personhood*.

Today's main solution is *email certification*. More precisely, when they create a Tournesol account, contributors are asked to validate, if possible, an email address from a trusted email domain. The list of trusted email domains is currently managed manually. An email domain will be considered trusted if it seems sufficiently unlikely that a large number of fake accounts can be created from this domain.

This excludes domains like @gmail.com and personal domains like @my-personal-website.com. The concern is not only that the domain will maliciously create a large number of fake accounts; it is also that they may be hacked by a malicious entity that will create such fake accounts. The list of trusted email domains is available at `https://tournesol.app/about/trusted_domains`. It includes domains like @epfl.ch, @who.int and @rsf.org. 795 contributors are thereby authenticated.

Evidently, however, this solution is still highly imperfect. On one hand, this does not guarantee the absence of fake accounts. On the other hand, and perhaps more importantly, this excludes most potential contributors from participating.

**Vouching mechanism.** To propagate trust to more accounts, Tournesol also proposes a vouching mechanism. Namely, any account can vouch for the authenticity of another account. More precisely, the account must vouch that the other account is used by a human who is not using any other account on the platform. The dataset contains 129 vouches.

**Comparison-based judgments.** Following a large literature on the topic [41, 19, 68, 11, 75, 64, 45], Tournesol relies on a comparison-based preference elicitation system. We believe that the need to distinguish among top content which should be more recommended makes this system more suitable than, e.g., using direct assessments [67, 2, 59, 88], which may yield too many "saturated" maximal assessments. Additionally, comparisons are labeled with the week in which the comparison was first submitted. This allows potentially observing changes or drifts in the contributors' judgments.

Figure 1 (left) presents the video comparison interface. Namely, contributors are asked to select two videos, and to tell Tournesol which one of the videos should be recommended at scale. Moreover, rather than a binary decision, the contributor is asked to provide the judgment by moving a slider on a more continuous scale, from $-10$ to $10$, The value $-10$ means that the contributor would prefer Tournesol to recommend the left video vastly more often than the right videos, while the value $0$ means that they believe both videos should be recommended equally often.

**Quality criteria.** Tournesol allows contributors to rate nine other *optional* quality criteria (Figure 1)

- **Reliable and not misleading:** Is the presented information trustworthy, robustly backed and properly nuanced?
- **Clear and pedagogical:** How efficiently does the content guide viewers in their understanding?
- **Important and actionable:** Can additional focus on this topic have a significantly positive impact on the world?
- **Layman-friendly:** How understandable is it, without prior knowledge?

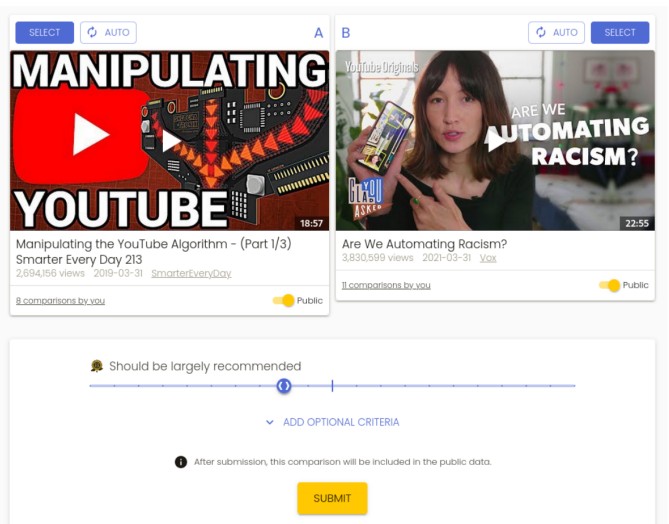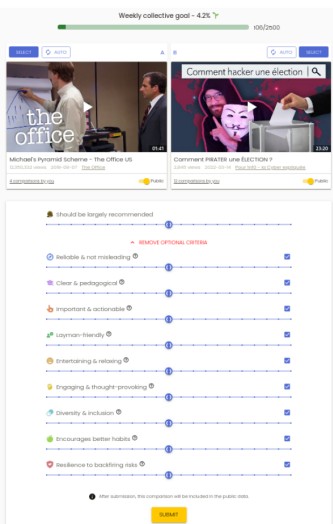

Figure 1: The interface through which contributors are asked to provide judgments. The judgments are comparisons of video contents using a slider along the main criteria "should be largely recommended" (left) and optional quality criteria (right).

- **Entertaining and relaxing:** Do people feel good watching it?
- **Engaging and thought-provoking:** Does it catch people's attention, spark curiosity and invite to question previous beliefs?
- **Diversity and inclusion:** Does it promote tolerance, compassion and wider moral considerations?
- **Encourages better habits:** Does it make people adopt habits that benefit themselves and beyond?
- **Resilience to backfiring risks:** Is it adapted to viewers with opposing beliefs? Does it prevent misconceptions or undesirable reactions?

While the criteria are further provided on Tournesol[2], most contributors have surely *not* read thoroughly our descriptions. Arguably, they will more likely judge these criteria according to their own understanding, which will be mostly based on the name of the criteria.

## 2.2 Processed data

In addition to the raw data presented thus far, the Tournesol public dataset exports processed data. The processing is performed by a pipeline called SOLIDAGO [14].

**Solidago.** The pipeline has six modules. First, pretrust and vouches are used to assign *trust scores* to all users. Second, *voting rights* are assigned to the different users, in a way that includes untrusted users, while guaranteeing that they cannot outweigh trusted users. Third, for each criterion and each user, the comparisons are turned into the user's *raw scores*, using the generalized Bradley-Terry model [36]. Fourth, raw scores are *scaled*, using Mehestan [4], zero-shift and standardization. Fifth, scaled scores are securely aggregated into *global scores*, using the Lipschitz-resilient quadratically regularized quantile [14]. Sixth, all scores are squashed into $(-100, 100)$, using the map $t \mapsto 100t/\sqrt{1 + t^2}$. All along, left and right uncertainties on all variables are computed.

**Exported values.** Trust scores, squashed individual scores and squashed global scores are provided in the public dataset.

**Results.** Figure 2 lists the most recommendable videos, according to Tournesol's contributors, as they are displayed on the website.

---

[2]https://tournesol.app/criteria

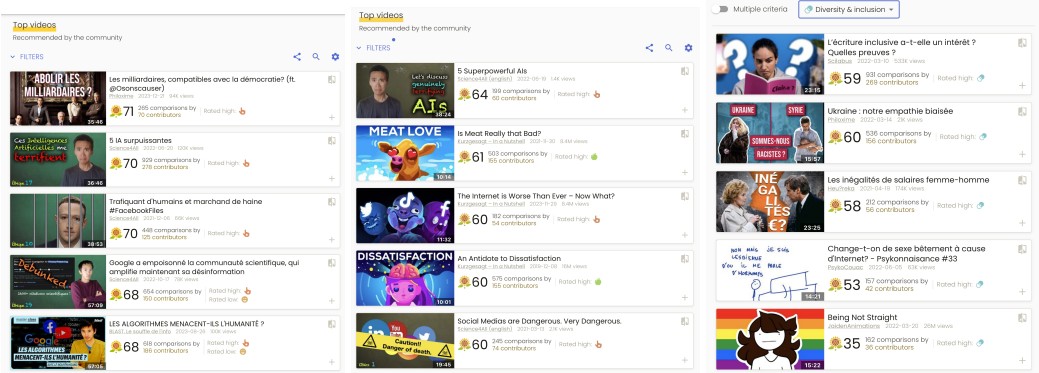

Figure 2: Best videos (left), best English-speaking videos (middle) and best videos along the criterion "diversity & inclusivity" (right).

## 2.3 Privacy

Overall, we encourage transparency in our contributors, as we believe that this will foster important research on human judgments, and help make safer and more ethical AIs. However, we acknowledge that, because of social and political pressures, some judgments are dangerous to make public, e.g. when criticizing one's own employer or government. This is why we allow contributors to provide data publicly or privately. More precisely, each contributor can select the privacy setting of any video they rate. If a video is rated privately, then all its comparisons to any other video will be recorded privately. Only Tournesol's server can access to such data. Conversely, all comparisons that involve two publicly rated videos are exported in the Tournesol public dataset.

## 2.4 Data collection context

The contributors to Tournesol receive no financial compensation. Their contributions are mostly motivated by the desire to contribute to a democratic AI governance project, and by the will to promote content of public interest. Their recruitment is thus organic, and mostly depends on how frequently they were exposed to the promotion of the Tournesol project. Evidently, this greatly correlates with Tournesol's communication, which has been heavily supported by the (French-speaking) YouTube channel Science4All, and by other science communicators [55]. As a result, the set of contributors is in no way representative of the global population. Namely, it is heavily biased towards science enthusiasts. Nevertheless, we believe that the data provided by this community should be of great interest to AI alignment, at least on topics with a significant scientific component.

# 3 Data analysis

This section presents some data analyses to provide insights in the *Tournesol public dataset*.

## 3.1 Contributors' contributions

Figure 3 displays the number of contributions per user. Perhaps unsurprisingly, this statistics is heavy-tailed; in fact, it seems to fit Zipf's law [85], with a few contributors providing most of the comparisons, and most of them providing very few. Figure 4 plots the activity through time: Tournesol has 100 to 200 weekly active users, while the number of monthly active users fluctuates between 200 and 900.

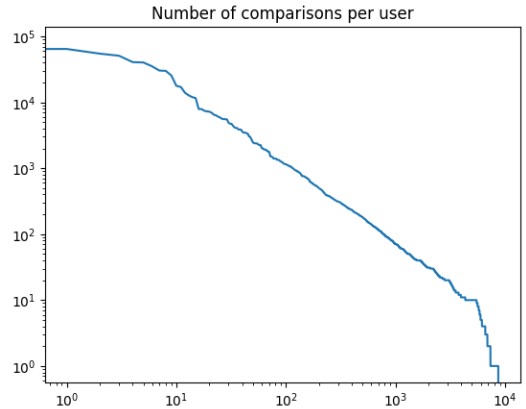

Figure 3: Number of comparisons provided by the different contributors, on a log-log scale, which is typical of Zipf's law [85].

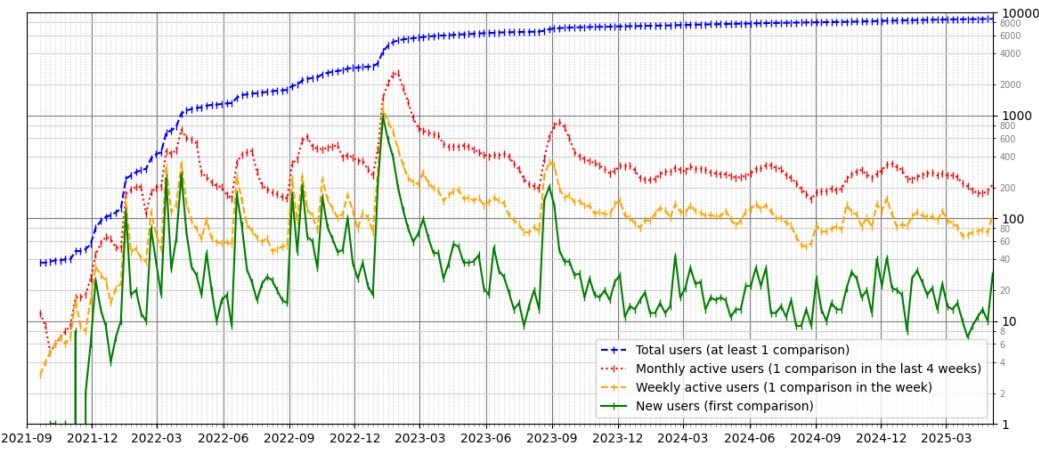

Figure 4: Contributors' participation through time.

## 3.2 Video and contributor connectivity

For scores to be meaningful, the contributors must have compared sufficiently many videos in common [4]. The contributor comparability graph has a connected component with 8,064 contributors and diameter of 6, out of the 8,692 contributors that have compared at least 2 videos. The graph has 256,360 edges out of 37,771,086 possible (0.68%) making it very sparse. But for the induced graph of the top 100 most active contributors with a trust at least 0.1 (which correspond to *scaling-calibration* contributors [14]), 3,699 (75%) pairs of contributors are comparable. This justifies the restriction of scaling calibration to the most active contributors.

Figure 5 details video comparisons for some highly active users. Interestingly, because the platform lets contributors to select their videos to compare, we observe a wide variety of comparison graphs. This raises open questions about the uncertainties of the resulting learned scores [36], and about the possibility to improve accuracy through *active learning* [69, 86].

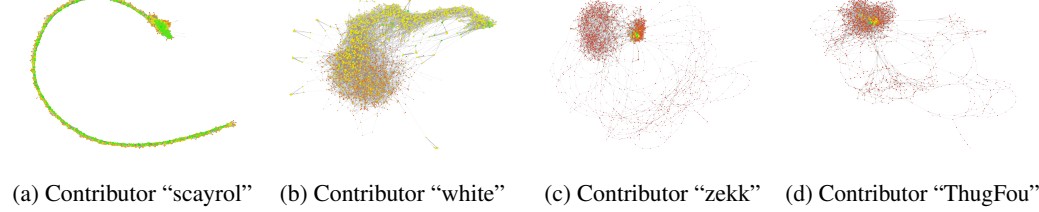

(a) Contributor "scayrol"    (b) Contributor "white"    (c) Contributor "zekk"    (d) Contributor "ThugFou"

Figure 5: Graphs of video comparisons for different users

## 3.3 Correlations between criteria

Figure 6 reports the correlations between quality criteria, in contributors' comparative judgments. Perhaps most remarkably, we observe that the criterion that best predicts whether a video "should be more largely recommended" is whether it is "important and actionable". This finding highlights the need to pay greater attention to *information prioritization*, and especially combatting "*mute news*" [55]. In particular, there may be an excess of attention to "*fake news*". In fact, [77] expose numerous strategies from the "merchants of doubts" that do not involve producing false information, such as shifting blame, cyberbullying critics or "striking a positive tone" [27].

Figure 6 also shows that most criteria are only weakly correlated. Two notable exceptions are "important and actionable" and "encourage better habits", and "reliable and not misleading" and "clear and pedagogical", which could be argued to be slightly redundant.

Note also that, as expected given Berkson's paradox [13], the correlations decrease if we only consider the top 10% videos on Tournesol (i.e. those that are more likely to be recommended).

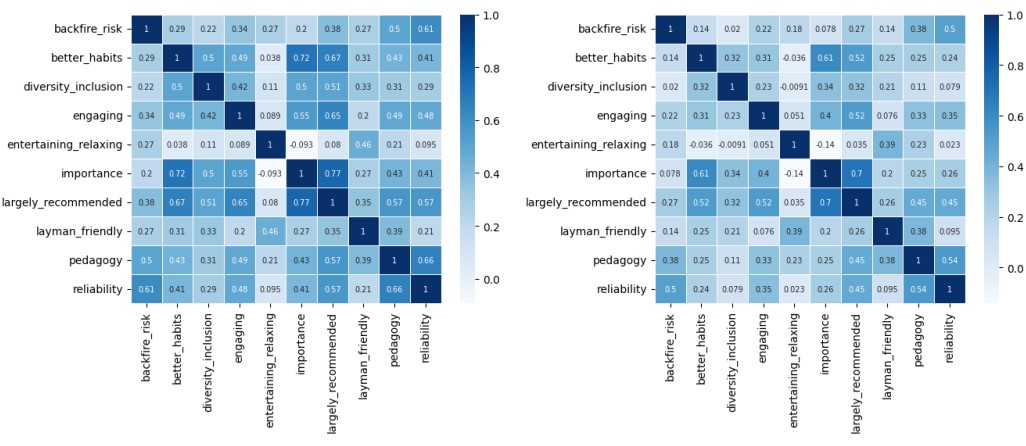

(a) All videos comparisons        (b) Comparisons between top 10% videos on Tournesol

Figure 6: Correlations between quality criteria

## 3.4 Distributions of reported comparisons

As it is not formally defined how contributors should rate a pair of videos, we expected many different expression styles. We ran a clustering algorithm (K-means) on statistics of the distribution of comparison values for each user. Figure 7 shows the typical distribution of comparison values of each of the eight clusters we identified. While some contributors provided comparisons close to "recommend equally" (cluster 3 and 4), others' comparisons were systematically towards the extreme (clusters 2, 5 and 6). This suggests that the discrepancies between their individual scores will be due to their expression style, rather than actual differences in their judgments, which justifies the research on mitigating the heterogeneity in expression styles [57, 96, 4].

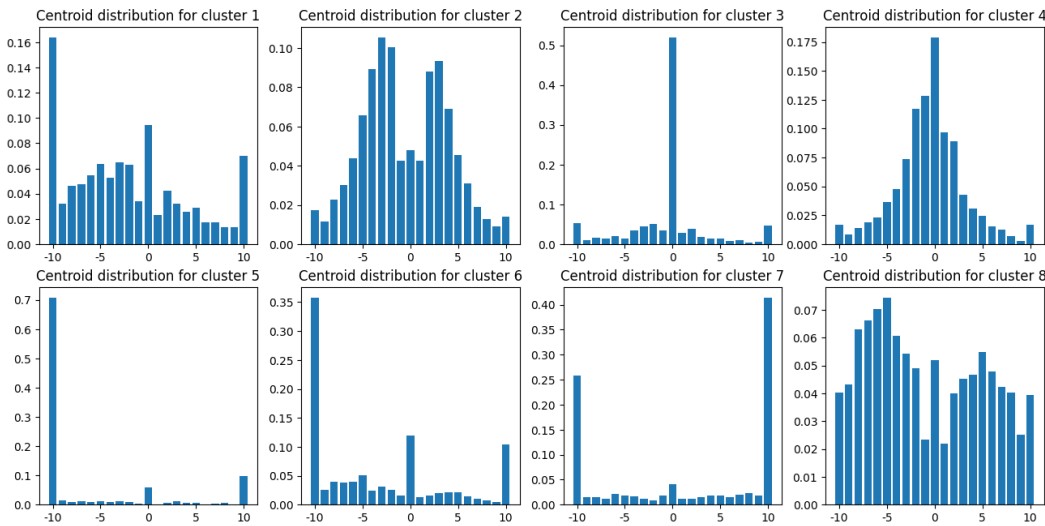

Figure 7: Example centroids of 8 clusters obtained by the K-means algorithm applied to the distributions of comparison values for each contributor with at least 20 comparisons. The clusters have sizes 144, 209, 47, 110, 23, 47, 42, 199.

## 3.5 Psychological biases in contributors' judgments

Our dataset exposes psychological biases in contributors' judgments. One example is a instinctive desire to over-recommend a recently watched high-quality video, known as the *recency bias* [66], which is depicted by Figure 8a. Namely, this figure plots all comparisons on the main criterion that correspond to a contributor evaluating a given video for the first time (negative scores correspond

to the newly scored videos). The 95% confidence interval for the mean of first-time comparisons is $[-0.39, -0.32]$, which is arguably a surprisingly significant bias.

Another bias we observe is a tendency to favor left videos. The 95% confidence interval for the mean of the main-criterion comparisons (Figure 8b) is $[-0.54, -0.5]$. Considering all criteria (Figure 8c) yields a smaller bias, with a corresponding 95% confidence interval of $[-0.19, -0.17]$. This suggests that reflecting on more criteria reduces the left-video bias. And indeed, when they are accompanied with comparisons on other criteria, the main-criterion comparisons have a 95% confidence interval for the mean equal to $[-0.38, -0.32]$, as opposed to $[-0.67, 0.61]$ for main-criterion-only comparisons. We also observe that pretrusted contributors have a significantly reduced left-video bias (on all criteria, $[-0.05, -0.03]$ for pretrusted, $[-0.35, -0.32]$ for unpretrusted).

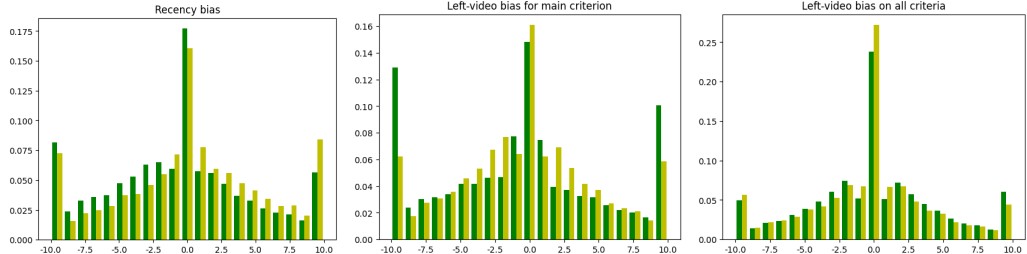

(a) First comparisons on main crite-(b) Comparisons on main criterion, (c) Comparisons on all criteria, sep-
rion (newly compared video is left). separated based on optional criteria. arated based on trust.

Figure 8: Recency and left-video biases in contributors' judgments.

## 3.6 Distribution of scores

Unsquashed scores (essentially, as outputs of the generalized Bradley-Terry model on contributors' comparisons) are extremely heavy tailed. Indeed, out of 791,264 scores, 4,581 deviate by more than 5 standard deviations. This is to be contrasted with the expected number 0.18 of such extreme scores, assuming a normal distribution of the scores. In fact, 428 scores deviate by more than 10 standard deviations. This observation justifies the use of comparisons to quantify the potential large deviations between top alternatives, which direct scoring approaches might fail to account for appropriately, as well as of a (robustified) quantile to standardize scores [14].

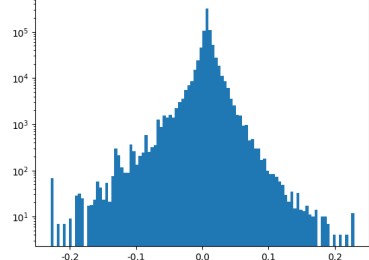

Figure 9: Distribution of un-squashed scores, with logarithmic y-scale.

## 4 Research challenges

Tournesol raises numerous fascinating research challenges. Below, we sketch some of these.

**Aggregate the different criteria into a score.** We expect the combination of many different quality criteria to yield a more reliable judgment of what content ought to be recommended at scale, or to a given specific user. However, the appropriate aggregation of our different quality criteria is still unclear, especially given probable nonlinear phenomena. How best to do this should be investigated.

**Debias the contributing population.** Like in many online participatory projects [10], we expect huge participation imbalances. Leveraging demographic data to debias the Tournesol recommendations, e.g., by giving more voting rights to individuals from underrepresented communities, could help, but it will require both (safely) collecting personal data and building new (secure) AIs, akin to those used by *Community Notes*[3] or by *Pol.is*[4].

**Volition.** As Section 3.5 highlighted, we cannot expect the Tournesol database to contain fully reliable human judgments. Many comparisons have surely been provided by contributors, at moments

---

[3]https://communitynotes.twitter.com/guide/en/under-the-hood/ranking-notes
[4]https://compdemocracy.org/algorithms/

when they were not paying the utmost attention to all the possible ramifications and unwanted side effects of promoting a video at scale. In particular, some judgments will arguably be more reliable than others. Such more reliable judgments are sometimes called *volitions*, rather than *preferences*. There is a need for AIs that model human psychology to distinguish between these two [54, 63].

**Privacy.** Tournesol's current AIs do not provide any *differential privacy* [29]. Future research should also investigate how to strengthen privacy without harming too much the quality and the security of the Tournesol scores. Perhaps most importantly, ideally, Tournesol's servers would be able to leverage private comparisons to score videos without being a single point of failure for private data protection. *Secure multi-party computations* could be a promising venue to do so [22].

**Decentralize Tournesol.** A longer-term goal is to fully decentralize Tournesol. In this vision, the data would no longer be stored on Tournesol's server, but would be replicated appropriately on a large number of contributors' devices. Moreover, the computations of Tournesol scores should also be decentralized, while guaranteeing *Byzantine resilience* [62]. Recent research in fully decentralized Byzantine learning has provided the building blocks of such a decentralization [32, 38], but more research is needed to understand how best to do so in the context of Tournesol.

**Preference generalization.** Right now, contributors are only voting on the videos that they explicitly compared. However, if they consistently voted positively all the videos of a given channel, then we could guess that they would have voted positively a new video from this channel, and to include their likely vote even when they did not compare the new video. Evidently, additional information can be leveraged to make such generalizations, such as the other video features (description, transcript, length), and the other contributors' judgments (using collaborative filtering [87]). Note however that generalization increases vulnerability risks. A careful security analysis would be required [70].

**Language model alignment.** Tournesol's database could help align language models, e.g. through *reinforcement learning with Tournesol feedback* [24, 79]. Determining how to combine large language models [40] with Tournesol's database to design safer models is an exciting venue for future work.

**Leverage expertise.** On technical topics like vaccination or climate change, especially when misconceptions are widespread in the general population, it seems desirable to assign more voting rights to experts, especially when judging the reliability of content within their domains of expertise. This issue is intimately connected to Condorcet's jury problem [25, 74].

**Proof of Personhood with zero knowledge.** Combatting fake accounts arguably remains the top priority to secure participatory systems. To address this, at least in democratic countries and in the short term, the state could be tasked with delivering *Proofs of Personhood* [18, 44], if possible in a zero-knowledge manner. More precisely, any citizen should ideally be able to provide to any platform a proof of citizenship, which does not enable neither the platform nor the state to identify which account is owned by which citizen. We believe that designing such a system could have applications beyond the particular case of Tournesol. Indeed, we could demand that social media only display the number of likes from users with a delivered proof of citizenship, and that their recommendation AIs be trained only by such certified users' data.

**Liquid democracy** Finally, future work could investigate the extent to which a liquid democracy [51] could be set up on plateforms like Tournesol. Such a system through which a contributor can delegate their votes to other voters could help combat activity bias (i.e. better accounting for inactive contributors) and expertise (if voters delegate to more competent contributors). While philosophically appealing, the security of such a system should however be first investigated [5].

## 5 Conclusion

This paper introduced the *Tournesol public dataset*, which is a large, secured and trustworthy database of reliable human judgments. We detailed its construction, and provided an analysis of its content. We believe that this database can help stimulate and facilitate research and development on ethical AIs, and could eventually help improve the informational diet of billions of people for the better. Given the current information crisis, we regard this as an "important and actionable" contribution.

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

# A    Datasheet for the Tournesol dataset

In this appendix, we provide a datasheet for the Tournesol dataset, based on the framework proposed by [47].

## A.1    Motivation

**For what purpose was the dataset created?**    The dataset was created to identify videos of public interest that should be recommended more largely. Additionally, we hope that the dataset will help motivate research on the ethics and security of recommendation algorithms.

**Who created the dataset and on behalf of which entity?**    The dataset was created by the nonprofit Tournesol Association, which is based in Switzerland.

**Who funded the creation of the dataset?**    The Tournesol Association is supporting the creation and maintenance of the dataset. It is in majority funded by crowdsourced donations, with occasional services to private companies.

## A.2    Composition

**What do the instances that comprise the dataset represent?**    The dataset contains mostly pairwise comparisons of videos by users. The dataset also contains vouches between users, authentication status, as well as processed data from this raw data.

**How many instances are there in total?**    The dataset contains 20k users (703 pretrusted), 40k videos, 126 vouches, 204k comparisons along the main criterion and 703k comparisons along optional criteria.

**Does the dataset contain all possible instances or is it a sample of instances of a larger set?**    The dataset contains all *public* judgments provided on the Tournesol platform.

**What data does each instance consist of?**    Each user has a pretrust status, based on email domain Sybil resilience. Each comparison is along a criterion, and refers to a user and a pair of videos.

**Is there a label or target associated with each instance?**    Each comparison takes a value between -10 and 10.

**Is any information missing from individual instances?**    Yes, plenty, such as the time it took to provide an answer, whether it was provided on a phone or a desktop, or whether the contributor actually watched the compared videos.

**Are relationships between individual instances made explicit?**    Some of them, yes, such as the contributor's identifier, or the videos that are compared.

**Are there recommended data splits?**    Yes, comparisons are naturally split by criterion, or by users. Trusted/untrusted contributions could be split.

**Are there any errors, sources of noise, or redundancies in the dataset?**    The comparisons come from humans, and are thus noisy, as well as potentially biased as discussed in the main part of the paper. Note that 4,446 comparisons were made before January 11, 2021, but because of a migration of the code, are dated on the January 11, 2021 week.

**Is the dataset self-contained, or does it link to or otherwise rely on external sources?**    The dataset refers to YouTube videos, but could be analyzed without knowledge of the videos.

**Does the dataset contain data that might be considered confidential?**    No. It was designed to be public.

**Does the dataset contain data that, if viewed directly, might be offensive, insulting, threatening or might otherwise cause anxiety?** Some poorly scored videos could be of this sort. Their content is not directly in the dataset, but the dataset points to them.

**Does the dataset identify any subpopulations?** Yes, trusted and untrusted contributors.

**Is it possible to identify individuals, either directly or indirectly, from the dataset?** Yes, especially given their public usernames.

**Does the dataset contain data that might be considered sensitive in any way?** Yes, indirectly, as it reveals consumption habits of contributors.

**Any other comments?** The individuals not only gave their consent, but the Tournesol also aims to make it clear that their provided data are used to design a democratic governance, and as such, could and should be scrutinized.

### A.3 Collection process

**How was the data associated with each instance acquired?** Through the Tournesol platform https://tournesol.app.

**What mechanisms or procedures were used to collect the data?** Through the Tournesol comparison interface https://tournesol.app/comparison.

**If the dataset is a sample from a larger set, what was the sampling strategy?** Based on public/private settings selected by the contributor.

**Who was involved in the data collection process and how were they compensated?** Contributors are volunteers, most of whom are recruited through promotion in science YouTube videos. They are not compensated.

**Over what timeframe was the data collected?** The first data was collected in May 2020. The collection has been continuously ongoing since.

**Were any ethical review processes conducted?** Not by an institutional review board, as our work was done by a nonprofit association.

**Did you collect the data from the individuals in question directly, or obtain it via third parties or other sources?** Yes, through the Tournesol platform that we designed.

**Were the individuals in question notified about the data collection?** Yes. They had to create a Tournesol account, to consent with the data collection, and to select whether to make their contributions public or not.

**Did the individuals in question consent to the collection and use of their data?** Yes.

**If consent was obtained, were the consenting individuals provided with a mechanism to revoke their consent in the future or for certain uses?** Yes, contributors can delete their Tournesol account, which will delete their data from Tournesol's (public) dataset.

**Has an analysis of the potential impact of the dataset and its use on data subjects been conducted?** Yes, we are consistently trying to make our project robustly beneficial.

### A.4 Preprocessing/cleaning/labeling

**Was any preprocessing/cleaning/labeling of the data done?** Yes. To output trust scores, as well as squashed individual and global scores.

**Was the "raw" data saved in addition to the preprocessed/cleaned/labeled data?** Yes. It is published in the Tournesol dataset.

**Is the software that was used to preprocess/clean/label the data available?** Yes. It is the open-source free-license Solidago python package.

### A.5 Uses

**Has the dataset been used for any tasks already?** Yes, it is used to make content recommendations to 10k+ users.

**Is there a repository that links to any or all papers or systems that use the dataset?** Such papers and systems are listed in `tournesol.app/#research`.

**What (other) tasks could the dataset be used for?**

**Is there anything about the composition of the dataset or the way it was collected and preprocessed/cleaned/labeled that might impact future uses?**

**Are there tasks for which the dataset should not be used?** The dataset should not be used to harm individuals, communities or society.

### A.6 Distribution

**Will the dataset be distributed to third parties outside of the entity (e.g., company, institution, organization) on behalf of which the dataset was created?** Yes. It is published on `api.tournesol.app/exports/all`.

**How will the dataset be distributed?** zip file downloadable from the website.

**When will the dataset be distributed?** Already is.

**Will the dataset be distributed under a copyright or other intellectual property license, and/or under applicable terms of use?** Yes, it is under ODC-By license.

**Have any third parties imposed IP-based or other restrictions on the data associated with the instances?** No.

**Do any export controls or other regulatory restrictions apply to the dataset or to individual instances?** Not to our knowledge.

### A.7 Maintenance

**Who will be supporting/hosting/maintaining the dataset?** The Tournesol association.

**How can the owner/curator/manager of the dataset be contacted?** `hello@tournesol.app`

**Is there an erratum?** No.

**Will the dataset be updated?** Yes. It is weekly updated, based on Tournesol's users newly reported data.

**If the dataset relates to people, are there applicable limits on the retention of the data associated with the instances?** No limit applies.

**Will older versions of the dataset continue to be supported/hosted/maintained?** Yes, the dataset is consistently updated every week, based on contributors' activity.

**If others want to extend/augment/build on/contribute to the dataset, is there a mechanism for them to do so?** The dataset is fully under the control of the Tournesol association. It is however under ODC-By license, thus any reuse is welcome, as long as attribution is appropriately provided.

