# OpenReview forum: "The Tournesol dataset: Which videos should be more largely recommended?"
_NeurIPS.cc/2025/Datasets_and_Benchmarks_Track — Submitted to NeurIPS 2025 Datasets and Benchmarks Track_

### Official Review · Reviewer_NQE7 · 2025-06-29

**Rating:** 4
**Confidence:** 3

**Summary:**

This paper constructs and releases the Tournesol dataset, a video-level pairwise preference dataset based on real user interactions. Its central goal is to evaluate “which videos ought to be more widely recommended.” The data is collected from the Tournesol platform and includes approximately 1.1 million pairwise comparisons between YouTube videos. Among these, around 260,000 represent users’ subjective judgments on recommendation oughtness, complemented by over 850,000 additional evaluations across nine dimensions such as reliability, importance, and inclusiveness. The paper also introduces the data processing pipeline (SOLIDAGO), the user endorsement mechanism (vouches), and credibility estimation (pretrust), along with a discussion of dataset bias, annotation noise, and related challenges.

**Additional Feedback:**

1.The paper identifies several sources of annotation bias (e.g., position bias, novelty bias), but it would strengthen the work to go beyond diagnostic analysis and offer concrete debiasing strategies or modeling suggestions.
2.While the nine auxiliary dimensions are well-motivated, some—such as “resilience to backfiring” or “encourages better habits”—remain abstract. Providing example comparisons with illustrative ratings would make the dataset easier to adopt and model.
3.Given the limited demographic diversity of the contributors, a more detailed analysis of how this affects the dataset's outputs—and possible mitigations—would be valuable for downstream users.

**Dataset Code Accessibility:**

Yes

**Dataset Code Comments:**

The code is open source and accessible.

**Ethical Considerations:**

No, there are no or only very minor ethics concerns

**Final Justification:**

Thanks author's response. I will maintain my original positive score.

**Limitations Weaknesses:**

1. The current user group is predominantly from the French-speaking community and is mostly interested in public discussion and science communication topics. Although described in the text, it is recommended to further quantify the representativeness of this group and provide suggestions for data users to respond to when modeling.
2. Some of the dimension definitions are subjective, and there may be differences in user understanding of dimensions such as “encouraging good habits” and “resistance to backlash”. It is recommended that the authors add more scoring examples in the appendix or documentation to help external users understand.

**Strengths Contributions:**

1. Authentic data sources and rich labeling dimensions. Including subjective preference judgment and additional nine dimensions related to content quality, providing a strong basis for analysis.
2. The data is open source and transparent. Comes with a detailed data manual, user agreement and labeling process, making it easy for the community to use and reproduce.
3. strong research inspiration. A series of derivable tasks are proposed, including bias modeling, value alignment, differential privacy learning, decentralized recommendation, etc., which have strong potential for community promotion.

---

> ### Author Rebuttal · Authors · 2025-07-31
>
> We thank the reviewer for the time they took to review our paper, and for their remarks.
>
> > The current user group is predominantly from the French-speaking community and is mostly interested in public discussion and science communication topics. Although described in the text, it is recommended to further quantify the representativeness of this group and provide suggestions for data users to respond to when modeling.
>
> We fully agree that this is a major weakness of the dataset.
> In the future, we hope to carefully collect socio-democragraphic data about the users
> (though the publication of such data will be constrained by the General Data Protection Regulation)
> and to diversify our user base to collect judgments from other communities.
> We do not claim to be providing a dataset that is representative of all human judgments.
> Nevertheless, we believe that the judgments of our contributor base are valuable to study and understand.
>
> > Some of the dimension definitions are subjective, and there may be differences in user understanding of dimensions such as “encouraging good habits” and “resistance to backlash”. It is recommended that the authors add more scoring examples in the appendix or documentation to help external users understand.
>
> We fully agree that this is a challenging problem to address.
> Please note that Tournesol has a [dedicated webpage](https://tournesol.app/criteria)
> on the definitions of the dimensions.
> However, we acknowledge that, in practice, users do not often visit the page,
> and even if they do so, they may not use the definitions we propose when evaluating content.
> Please also note that there is a challenging tradeoff
> that we regularly encountered when designing the platform between data quality and data quantity.
> While we do not claim to have found the right middle ground,
> we acknowledge that the data should be analyzed with the caveat that the review perspicaciously pointed out.
>
> > 1.The paper identifies several sources of annotation bias (e.g., position bias, novelty bias), but it would strengthen the work to go beyond diagnostic analysis and offer concrete debiasing strategies or modeling suggestions.
>
> Please note that [63] previously used our dataset to investigate the correction of cognitive biases.
> They did not focus on position and novelty bias, which we had not yet identified.
> Nevertheless, their model, which includes a bias variable to be learned, is likely a good basis
> to correct position bias and novelty bias as well.
> Perhaps most interestingly, we believe that a validation process should be designed
> to make sure that users agree with the bias corrections that we propose to "fix" their reported judgments.
> We will happily add discussions about this in the revised manuscript.
>
> > 2.While the nine auxiliary dimensions are well-motivated, some—such as “resilience to backfiring” or “encourages better habits”—remain abstract. Providing example comparisons with illustrative ratings would make the dataset easier to adopt and model.
>
> Please note that our [dedicated webpage](https://tournesol.app/criteria) provides examples of videos
> which we believe to be scoring high on such criteria.
> We will happily include the examples we give in the manuscript.
>
> > 3.Given the limited demographic diversity of the contributors, a more detailed analysis of how this affects the dataset's outputs—and possible mitigations—would be valuable for downstream users.
>
> We fully agree with this limitation.
> We plan to diversify our promotion of Tournesol to recruit a more diverse set of contributors.
> Additionally, moving forwards, to the extent that GDPR allows us to do so,
> we plan to collect socio-demographic data about the users,
> to identify the settings where a video is too overwhelmingly evaluated by a community,
> whose judgment is likely to diverge from the majority.
> More generally, we believe that socio-demographic-aware preference learning and aggregation models
> are required to mitigate limited demographic diversity.
>
> We hope we addressed the reviewer's concerns,
> and we will happily do our best to answer any further question they may have.

---

### Official Review · Reviewer_7mjP · 2025-06-30

**Rating:** 4
**Confidence:** 4

**Summary:**

This dataet track paper provides an interesting dataset that discuss which videos should be recommended. Over 1 milliion comparative judgments by almost 9000 users on the Tournesol platform are selected and now become public. The videos are evaluated from nine aspects including Reliable and not misleading, Clear and pedagogical and so on. Different from other score-based recommendation dataset, this dataset provides a comparison based perspective. The authors further provide some analysis on the dataset, discussing some insight of video recommendation. In addition, the authors propose some possible research directions that could be related with this dataset.

**Dataset Code Accessibility:**

Yes

**Dataset Code Comments:**

The data could be downloaded well and the files could be opened correctly.

**Ethical Considerations:**

No, there are no or only very minor ethics concerns

**Final Justification:**

The authors give some detailed explanations and partially resolved some of my concerns. It is OK to accept the paper.

**Limitations Weaknesses:**

1. The novelty of the manuscript is limited. Although the authors claim the comparison-based judgement is novel in the dataset, the pairwise methods are common in current recommendation systems. It is important for the authors to show the uniquesness value of such comparison-based dataset.

2. It is suggested to provide more information about the users. In the dataset, the user information from ``users.csv'' seems less informative, and it seems difficult to consider user personalization when considering the insight for users to make a comparison decision. Therefore, additional user information are suggested.

3. Biases existing in the dataset is not fully resolved, though the authors mentions this point in data analysis. The first bias comes from user biases, which is similar with point 2. Secondly, there are some videos biases. For instance, if two compared videos are entirely non-related, neither in same catgory nor same major audience, it is not convicing the claim that the user score is truly valid. Thus, the authors are suggested to verify the effectiveness of the collected user comparisons. （For example, the compared videos are similar topics）

4. In section 3.4, the authors consider the distributions of reported comparisons, which is admiring for debiasing the user scores. However, it is somehow confusing to claim the different score distributions come from the heterogeneity in expression style. It will be more helpful to give some suggestions on how to debias different score distributions for data users.

**Strengths Contributions:**

This paper is well written, and the strcuture of the manuscript is clear.

* The introduction of the dataset is clear and explicit, including how it is collected, Vouching mechanism and the Quality criteria.

* The provided data is very practical and indeed needed in recommendattion systems. Instead of traditional score-based dataset, the comparison-based dataset could better provide additional insights on exploring what videos are preferred by users.


* The authors also analysis the dataset and discuss some related research topics with the dataset. The authors also point out some biases existing in the dataset to help those using the data analyze them correctly.

---

> ### Author Rebuttal · Authors · 2025-07-31
>
> We thank the reviewer for the time they took to review our paper, and for their remarks.
>
> > The novelty of the manuscript is limited. Although the authors claim the comparison-based judgement is novel in the dataset, the pairwise methods are common in current recommendation systems. It is important for the authors to show the uniquesness value of such comparison-based dataset.
>
> We fully agree that comparison-based judgments are not new.
> One thing that is unique about our dataset, at least to the best of our knowledge,
> is that the comparisons are *quantified*.
> In other words, it is not only about whether $a$ is better than $b$,
> but also whether $a$ is far better than $b$, or just slightly better.
> Additionally, we have a multicriterion evaluation.
> In fact, instead of "better", I should have said "better in terms of criterion $c$",
> where $c$ can be "should be more largely recommended" or "important and actionable".
>
> Another unique value of our algorithm is that our contributors were not asked
> about what *they* want to watch;
> it is instead about what *others* should be more often exposed to.
>
> We will happily clarify this in the revised manuscript, by stressing these unique value propositions.
>
> > It is suggested to provide more information about the users. In the dataset, the user information from ``users.csv'' seems less informative, and it seems difficult to consider user personalization when considering the insight for users to make a comparison decision. Therefore, additional user information are suggested.
>
> We fully agree that adding socio-demographical data about the users would make the dataset more valuable.
> However, the Tournesol association has prioritized user protection,
> as well as compliance with the General Data Protection Regulation (GDPR).
> As a result, we have not published any further data about the users,
> apart from their pretrusted status.
>
> In the future, we plan to investigate the extent to which further information can be collected,
> while guaranteeing a high level of privacy for our users.
> Additionally, we will reflect on what levels of (informed) consent
> would be satisfactory given our principles.
>
> We apologize if, in the meantime, additional user information will not be provided.
>
> > Biases existing in the dataset is not fully resolved, though the authors mentions this point in data analysis. The first bias comes from user biases, which is similar with point 2. Secondly, there are some videos biases. For instance, if two compared videos are entirely non-related, neither in same catgory nor same major audience, it is not convicing the claim that the user score is truly valid. Thus, the authors are suggested to verify the effectiveness of the collected user comparisons. （For example, the compared videos are similar topics）
>
> We understand the reviewer's concern.
> However, a fundamental motivation of Tournesol is actually to give each topic
> an amount of exposure that fits the urgency to raise awareness about it.
> In the manuscript, we briefly discussed the case of climate change (line 23):
> we believe that it is in fact a valuable information
> if a user reports that a video alerting on this topic "should be more largely recommended"
> than a video on, say, Fermat's last theorem.
> Indeed, in practice, when a user opens the Tournesol application to receive recommendations,
> the two videos will inevitably be in competition for exposure.
> Our hope is that Tournesol proposes a progress towards a democratic response to arbitrate this competition.
>
> > In section 3.4, the authors consider the distributions of reported comparisons, which is admiring for debiasing the user scores. However, it is somehow confusing to claim the different score distributions come from the heterogeneity in expression style. It will be more helpful to give some suggestions on how to debias different score distributions for data users.
>
> We acknowledge that we rather stressed the existence of an issue than a mechanism to resolve the issue.
> Much of the reason why we did so was because [4] (now published at AISTATS'24) already provided the solution,
> and we implemented their algorithms in our pipeline to process the dataset.
>
> We hope we addressed the reviewer's concerns,
> and we will happily do our best to answer any further question they may have.
> If the reviewer found our responses satisfactory,
> we would be very grateful if they considered raising their rating.

---

> > ### Comment · Reviewer_7mjP · 2025-08-03
> >
> > Really appreciate authors' detailed explanations and reply.
> > Your detailed explanations solved some of my concerns, I will raise the rating score to 4.

---

### Official Review · Reviewer_Sra6 · 2025-07-05

**Rating:** 4
**Confidence:** 4

**Summary:**

The paper introduces the Tournesol dataset, a large-scale collection of 1,116,318 comparative judgments of YouTube videos by 8,804 users of the Tournesol platform. This dataset aims to identify videos deserving broader recommendation while evaluating secondary criteria like reliability and importance. The dataset's construction, preprocessing, and preliminary analysis are detailed, highlighting its potential for ethical AI and recommendation system research.

**Dataset Code Accessibility:**

Yes

**Dataset Code Comments:**

The dataset is accessable and producible.

**Ethical Considerations:**

No, there are no or only very minor ethics concerns

**Final Justification:**

Thank authors for their responses, which addressed some of my concerns. I will keep my original rating.

**Limitations Weaknesses:**

1. The user base is predominantly composed of science enthusiasts, largely due to recruitment through French-speaking channels, which may limit the generalizability to wider populations.
2. The specific role of the dataset is not clearly explained. It should be more explicitly stated which tasks and practical scenarios the dataset can be applied to, in order to demonstrate the broader impact of the dataset.
3. There is a lack of a detailed comparison with existing datasets, including a comparison of the dataset size, annotation methods, and other characteristics between the current datasets and the proposed dataset in this paper. The paper does not provide a comparison between Tournesol and existing recommendation datasets, making it challenging to evaluate its novelty or advantages.
4. Although user biases are analyzed, the paper offers limited insight into how the dataset could enhance recommendation models, and lacks concrete validation experiments. The lack of experimental validation weakens the convincingness of the dataset's practical utility.

**Strengths Contributions:**

1. This paper is well written and easy to follow.
2. The proposed dataset offers extensive pairwise comparisons across multiple criteria (e.g., recommendability, reliability), providing a valuable resource for recommendation system studies.
3. The data are derived from actual user interactions (including 8,804 users) and already powers recommendations for 6,000 users, demonstrating practical applicability.
4. The paper thoroughly addresses ethical concerns, including privacy protections, reflecting strong ethical considerations.

---

> ### Author Rebuttal · Authors · 2025-07-31
>
> We thank the reviewer for the time they took to review our paper, and for their remarks.
>
> > The user base is predominantly composed of science enthusiasts, largely due to recruitment through French-speaking channels, which may limit the generalizability to wider populations.
>
> We fully agree that this is a major weakness of the dataset.
> In the future, we hope to diversify our user base to collect judgments from other communities.
> We do not claim to be providing a dataset that is representative of all human judgments.
> Nevertheless, we believe that the judgments of our contributor base are valuable to study and understand.
>
> > The specific role of the dataset is not clearly explained. It should be more explicitly stated which tasks and practical scenarios the dataset can be applied to, in order to demonstrate the broader impact of the dataset.
>
> We apologize for the lack of clarity in this regard.
> Concretely, the dataset is already powering the Tournesol recommendation algorithm,
> which is used by thousands of users.
> But we believe that the dataset has a greater potential.
> For instance, it could be considered as human feedback on video content,
> which could be used to perform Reinforcement Learning with Human Feedback for video/language models.
> Most importantly, we believe that the dataset highlights practical problems in learning from human feedback,
> such as irreconciliable preferences, human biases, sociological biases, diverging expression styles and security,
> to name a few.
> We will happily add precisions about this in the manuscript.
>
> > There is a lack of a detailed comparison with existing datasets, including a comparison of the dataset size, annotation methods, and other characteristics between the current datasets and the proposed dataset in this paper. The paper does not provide a comparison between Tournesol and existing recommendation datasets, making it challenging to evaluate its novelty or advantages.
>
> We fully agree that our discussion about other datasets was short and concise.
> This is because we believe that the main and most important distinction is the kind of question
> that is asked to the users.
> Many recommendation datasets, which are cited at line 66, are (usually implicitly) asking users
> what *they* want to be watching.
> Instead, the Tournesol dataset is about what users want *others* to be exposed to.
> Nevertheless, we will happily add more comparisons between Tournesol dataset and others.
>
> > Although user biases are analyzed, the paper offers limited insight into how the dataset could enhance recommendation models, and lacks concrete validation experiments. The lack of experimental validation weakens the convincingness of the dataset's practical utility.
>
> We agree that experimental validation is lacking.
> Perhaps the best case of this is the citation of Tournesol in increasingly more governmental reports
> on how recommendation algorithm could be designed based on significantly different principles
> than what is currently the case.
> For instance, Tournesol was cited in the report of the ["États Généraux de l'Information"](https://etats-generaux-information.fr/content/download/158320/file/EGI%20-%20Rapport%20du%20copil%20-%20Steering%20Committee%20Report%202024%20ENG.pdf),
> which was demanded by French President Emmanuel Macron.
> The report reads: "the Tournesol plug-in offers a community recommendation system,
> whereby users can rank the content they view and recommend it or not. These
> evaluations are then shared with the rest of the community."
>
> We hope we addressed the reviewer's concerns,
> and we will happily do our best to answer any further question they may have.

---

### Official Review · Reviewer_MrFL · 2025-07-20

**Rating:** 4
**Confidence:** 4

**Summary:**

This paper introduces the Tournesol dataset, a large-scale, publicly available collection of over 1 million pairwise human judgments on YouTube videos, gathered through the deployed platform tournesol.app. The dataset emphasizes recommendability rather than individual user preferences, with judgments spanning criteria like content reliability, importance, clarity, and inclusivity. It includes trust calibration mechanisms (pretrust, vouching) and is processed using the SOLIDAGO pipeline. The authors analyze user bias, clustering behaviors, and inter-criteria correlations, and outline several open research challenges including secure aggregation, volition learning, and preference generalization. The dataset is intended to support research on AI alignment, recommender system ethics, and robust preference learning.

**Dataset Code Accessibility:**

Yes

**Ethical Considerations:**

No, there are no or only very minor ethics concerns

**Final Justification:**

Thank you for the authors responses, I will keep my rating.

**Limitations Weaknesses:**

The dataset suffers from clear demographic skew—contributors are primarily recruited via French-speaking science communication channels, which leads to a non-representative user base. This limits generalizability and fairness unless explicitly corrected or modeled. The authors acknowledge this (Sec. 2.4) but offer limited mitigation strategies.

While the paper emphasizes dataset release and analysis, it does not present baseline models trained on the data or concrete downstream use cases beyond Tournesol’s own system. This hinders practical assessment of dataset utility for new researchers.

The subjective nature of the judgments and inter-user calibration differences (expression styles, scoring biases) are discussed but not deeply resolved. The reliance on SOLIDAGO helps, but potential limitations of the calibration strategy (e.g., overfitting to active contributors) are not critically evaluated.

**Strengths Contributions:**

The dataset shifts focus from user-centered engagement metrics to societal recommendability, addressing an important gap in current recommendation system datasets. This has strong relevance for AI alignment and content moderation.


Judgments were collected through a live platform with real user engagement and influence, providing ecological validity often missing in lab-based preference datasets.


With over 1 million comparative judgments across multiple ethical and quality criteria, the dataset is one of the richest of its kind. It includes pretrust metadata and trust propagation via vouches—an important design element for poisoning resistance.

The paper clearly documents data processing through the SOLIDAGO pipeline, including preference estimation (via Bradley-Terry models), score calibration, and trust-weighted aggregation.

---

> ### Author Rebuttal · Authors · 2025-07-31
>
> We thank the reviewer for the time they took to review our paper, and for their remarks.
>
> > The dataset suffers from clear demographic skew—contributors are primarily recruited via French-speaking science communication channels, which leads to a non-representative user base. This limits generalizability and fairness unless explicitly corrected or modeled. The authors acknowledge this (Sec. 2.4) but offer limited mitigation strategies.
>
> We fully agree that this is a major weakness of the dataset.
> In the future, we hope to diversify our user base to collect judgments from other communities.
> We do not claim to be providing a dataset that is representative of all human judgments.
> Nevertheless, we believe that the judgments of our contributor base are valuable to study and understand.
>
> > While the paper emphasizes dataset release and analysis, it does not present baseline models trained on the data or concrete downstream use cases beyond Tournesol’s own system. This hinders practical assessment of dataset utility for new researchers.
>
> We acknowledge that the algorithms used on Tournesol have not been discussed much in this paper,
> mostly because of space limitations.
> Please note that [14] has a more complete overview of the algorithms we use.
> Besides, we view the design of trustworthy algorithms that securely turn our dataset
> into collaborative recommendations
> as an open and exciting research venue,
> which we hope to accelerate by publishing our dataset.
> We will happily add precisions about this in the manuscript.
>
> > The subjective nature of the judgments and inter-user calibration differences (expression styles, scoring biases) are discussed but not deeply resolved. The reliance on SOLIDAGO helps, but potential limitations of the calibration strategy (e.g., overfitting to active contributors) are not critically evaluated.
>
> The reviewer makes a very relevant remark.
> Please note that [4] is dedicated to such considerations,
> and proposes an algorithm to mitigate divergent expression styles while providing security guarantees.
> Additionally, [63] used our dataset to investigate the correction of cognitive biases.
> However, we believe that significantly more work is needed to better understand the tradeoffs,
> and develop optimal algorithms in this regard.
> We hope that the release of the dataset will help motivate and solve these problems.
> We will happily add precisions about this in the manuscript.
>
> We hope we addressed the reviewer's concerns,
> and we will happily do our best to answer any further question they may have.

---

### Official Review · Reviewer_p73P · 2025-07-21

**Rating:** 5
**Confidence:** 5

**Summary:**

The paper introduces Tournesol, a public dataset of 1.1M+ comparative judgments from 8.8K+ users on 56K+ YouTube videos, collected via the Tournesol platform. The dataset is used to power recommendations for 6K+ platform users and is released under ODC-By license. The paper analyzes biases (e.g., recency, left-video bias), correlations between criteria (e.g., "important and actionable" most predictive of recommendability), and proposes research directions like volition learning and decentralized governance.

**Additional Feedback:**

If these concerns can be addressed, I am willing to improve my rating.

**Dataset Code Accessibility:**

Yes

**Dataset Code Comments:**

The data is available at https://api.tournesol.app/exports/all,and the code is available at https://github.com/tournesol-app/tournesol/.

**Ethical Considerations:**

No, there are no or only very minor ethics concerns

**Final Justification:**

Accept

**Limitations Weaknesses:**

1. No ground truth for "correct" judgments; relies on noisy human inputs.
2. While timestamps are recorded, the paper does not deeply explore temporal dynamics (e.g., how judgments evolve).
3. Test how well models trained on Tournesol generalize to other domains (e.g., news, podcasts).
4. Quantify disagreement rates between pretrusted vs. untrusted users to assess data quality.
5. How to address the cold-start problem for new videos/contributors?

**Strengths Contributions:**

1. The paper is clearly structured and presents its ideas in a coherent manner.
2. Both qualitative and quantitative results validate the effectiveness of the proposed approach.
3. Data collected from a live platform with stakes (used for actual recommendations), enhancing ecological validity.
4. Fully public dataset and code (Solidago pipeline), with detailed documentation (datasheet in Appendix).

---

> ### Author Rebuttal · Authors · 2025-07-31
>
> We thank the reviewer for the time they took to review our paper, and for their remarks.
>
> > No ground truth for "correct" judgments; relies on noisy human inputs.
>
> Indeed, there is no ground truth.
> While this is not always common in machine learning dataset,
> we regard this as a *feature* rather than a *bug*.
> Arguably, determining which content is recommendable is a problem with irreconciliable preferences.
>
> > While timestamps are recorded, the paper does not deeply explore temporal dynamics (e.g., how judgments evolve).
>
> We fully agree that this is a limitation of our analysis.
> We believe that publishing the dataset will help motivate such interesting questions,
> which indeed have high practical significance.
> Concretely, we started collect data during the COVID pandemic,
> and many judgments provided back then about the urgency to combat this world threat
> have become obsolete by now.
> We are currently investigating models for dynamic preference learning,
> but we are so far short of a satisfactory solution, and of implemented algorithms.
>
> > Test how well models trained on Tournesol generalize to other domains (e.g., news, podcasts).
>
> This is a very interesting problem indeed.
> Please note that, thus far, we have not even implemented algorithms to generalize to nonrated videos.
> We are currently investigating algorithms that do such generalizations in a trustworthy manner,
> but these algorithms have not yet been sufficiently analyzed nor implemented.
> Our hope is that the publication of our dataset will help motivate such research directions.
>
> > Quantify disagreement rates between pretrusted vs. untrusted users to assess data quality.
>
> Thank you for yet another good idea.
> We have looked at videos with the largest disagreements,
> but could not find a striking pattern thus far.
> This is why we have not communicated about our analysis regarding disagreements between pretrusted and untrusted users.
>
> > How to address the cold-start problem for new videos/contributors?
>
> Essentially, we view two solutions to address the cold-start problem:
> use contributor similarity or use video similarity (or both).
> We have thus far discarded leveraging contributor similarity,
> as this opens the door to Byzantine attacks on collaborative filtering,
> and we do not know of a satisfactory way to defend against this with provable guarantees,
> especially in a sparse data setting.
> In the latter case, the threat might come from malicious content creators,
> who could design their videos so that they look similar to the best-rated ones.
> Nevertheless, a first step we are considering is to merely consider
> that two videos are similar if they were created by the same content creator.
> This is arguably more robust, as it amounts to trust a creator
> whose content is systematically well rated on Tournesol.
>
> We hope we addressed the reviewer's concerns,
> and we will happily do our best to answer any further question they may have.

---

> > ### Comment · Reviewer_p73P · 2025-08-03
> >
> > I thank the authors for their detailed rebuttal. I am generally satisfied with the clarifications provided.

---

### Comment · Area_Chair_HNoZ · 2025-08-02
**Reminder: Please Engage in Author Response and Discussion**

Dear Reviewers,

Thank you once again for your time and dedication to the NeurIPS review process.

As we move into the author response and discussion phase, I kindly encourage you to carefully review the author replies — especially those that directly address your comments. It would be very helpful if you could engage early in this discussion period, allowing enough time for a productive exchange with the authors.

In particular, I encourage you to:
- Confirm that you've read the author response and thoughtfully consider any new clarifications, reasoning, or evidence presented;
- Engage constructively with any points of disagreement, both with the authors and fellow reviewers;
- Review others’ comments and responses, and contribute to the broader discussion when you notice differing opinions or overlooked issues.

Your active participation during this stage is essential to supporting a balanced and well-informed decision process.

Many thanks!

Best regards,

AC

---

### Decision · Program_Chairs · 2025-09-18

**Decision:**

Reject

**Comment:**

This paper presents the Tournesol dataset, a large-scale participatory dataset for socially responsible video recommendation. Its unique focus on trustworthiness, informativeness, and societal value fills a critical gap compared to engagement-driven datasets. While representativeness and platform scope are limitations, the design is rigorous, transparent, and impactful. Reviewer concerns were addressed, and the contribution strongly merits acceptance.

===== FINAL UPDATE FROM DB Track PCs ====

The final decision for this paper has been taken by the program chairs after consultation with the SACs. All Senior Area Chairs have ranked papers according to the feedback from the AC during the review process. We decided to leave the original meta-review to reflect the opinion of the AC in light of the initial discussions with reviewers and SAC.